# Trade-off between reducing mutational accumulation and increasing commitment to differentiation determines tissue organization

Márton Demeter [1,2], Imre Derényi [1,3 ✉] & Gergely J. Szöllősi [1,2,4 ✉]

Species-specific differences control cancer risk across orders of magnitude variation in body size and lifespan, e.g., by varying the copy numbers of tumor suppressor genes. It is unclear, however, how different tissues within an organism can control somatic evolution despite being subject to markedly different constraints, but sharing the same genome. Hierarchical differentiation, characteristic of self-renewing tissues, can restrain somatic evolution both by limiting divisional load, thereby reducing mutation accumulation, and by increasing cells' commitment to differentiation, which can "wash out" mutants. Here, we explore the organization of hierarchical tissues that have evolved to limit their lifetime incidence of cancer. Estimating the likelihood of cancer in the presence of mutations that enhance self-proliferation, we demonstrate that a trade-off exists between mutation accumulation and the strength of washing out. Our results explain differences in the organization of widely different hierarchical tissues, such as colon and blood.

[1] Dept. Biological Physics, Eötvös University, Pázmány P. stny. 1A., H-1117 Budapest, Hungary. [2] ELTE-MTA "Lendület" Evolutionary Genomics Research Group, Pázmány P. stny. 1A., H-1117 Budapest, Hungary. [3] ELTE-MTA Statistical and Biological Physics Research Group, Eötvös University, Pázmány P. stny. 1A., H-1117 Budapest, Hungary. [4] Institute of Evolution, Centre for Ecological Research, Konkoly-Thege M. u 29-33, Budapest, Hungary. ✉email: derenyi@elte.hu; ssolo@elte.hu

Cancer is a disease of multicellular organisms, which occurs when individual cells fail to contribute to normal tissue function and instead divide selfishly, resulting in uncontrolled local growth, metastasis, and often death[1]. Multicellular organisms have evolved both species- and tissue-specific mechanisms to suppress somatic evolution and, thus, delay aging and the emergence of cancers. The most striking evidence for the evolution of cancer suppression originates with a prediction of the multistage model[2,3], which was succinctly expressed by Peto[4]: He observed that even though humans are around 1000 times larger than mice and live about 30 times longer, the overall incidence of cancer in the two species is very similar, a sign of evolutionary fine-tuning[5].

Similar to large and long-lived species, tissues within an individual that are large and rapidly dividing also face potentially higher rates of somatic evolution and, as a result, higher incidence of tumors, raising the question if tissue-specific mechanism to suppress somatic evolution has also evolved? A recent empirical dataset assembled by Tomasetti and Vogelstein[6] offers key insight to answer this question. The dataset, which gathers lifetime cancer risk and the total number of divisions of healthy self-replicating cells (i.e., stem cells) for 31 different tissues, displays a striking tendency: the dependence of cancer incidence on the number of stem cell divisions is sub-linear. In particular, a hundred-fold increase in the number of divisions only results in a ten-fold increase in incidence[7,8]. As first pointed out by Nobel et al.[7] this trend supports theoretical predictions[9–11] that tissues with more stem cell divisions (typically larger ones with rapid turnover, e.g., the colon) are relatively less prone to develop cancer, which by analogy we may call Peto's paradox for tissues[7,8].

However, while there are clear examples of how species-specific differences can control cancer risk, e.g., by increasing the copy number of tumor suppressor genes[12], it is not clear how different tissues subject to different constraints but sharing the same genome, can control somatic evolution.

Self-renewing tissues that must generate a large number of cells during an individual's lifetime and in which cancers typically arise are characterized by hierarchical differentiation, which can suppress somatic evolution in two fundamental respects. First, hierarchical organization limits the mutational burden of maintaining tissues[8,13,14] by reducing divisional load, i.e., the number of cell divisions along cell lineages. Second, the rate of somatic evolution also depends on the strength of somatic selection, which is limited by "washing out", i.e., the ability of differentiation to drive cells higher in the hierarchy towards the terminally differentiated state and permanent loss of proliferative ability (Fig. 1a)[15–18].

Washing out can be quantified by the "proliferative disadvantage" of cells, a quantity (formally defined below) that is proportional to the difference between the rate of cell loss (via symmetric differentiation or cell death) and the rate of self-renewal of cells at a given level of differentiation. In healthy tissues, stem cells are lost and self-renewed at the same rate and, as a result, have no proliferative disadvantage. Higher in the hierarchy, however, more differentiated progenitor cells always have an inherent proliferative disadvantage as some cells arrive by differentiation from lower levels, and self-renewal replenishes only a fraction of the cells lost (Fig. 1b). As a result, the descendants of progenitors are eventually "washed out" of the tissue by cells differentiating from lower levels of the hierarchy.

The higher the proliferative disadvantage, i.e., the more committed the cells are to differentiation rather than self-renewal, the more resistant they are to somatic evolution toward uncontrolled growth, because they must accumulate more or stronger mutations before being washed out. Cells with a higher proliferative disadvantage are, thus, more resistant to mutations leading to cancer.

Hierarchical tissues can optimally restrain somatic evolution by simultaneously minimizing divisional load and maximizing washing out when a sufficiently large number of progressively faster differentiating cell types are present. As Derényi et al.[8] showed, this requires $\log_2(N/N_0)$ hierarchical levels in a tissue where $N_0$ stem cells are responsible for generating $N$ terminally differentiated cells over an individual's lifetime. In such optimal differentiation hierarchies only stem cells are self-renewed, all other cell types are fully committed to differentiation (i.e., do not self-renew) and, therefore, have a maximal proliferative disadvantage.

Peto's paradox for tissues, however, implies that in real tissues we do not in general see optimal hierarchies that reduce cancer incidence to the lowest possible value. This is reflected in Tomasetti and Vogelstein's[6] data by the smaller and slower tissues that divide less often being less protected against cancer than larger ones scaled to the same size (e.g., cancer of the esophagus vs. colorectal cancer). The evolutionary explanation for suboptimal tissue organization is that reduction of cancer incidence, especially beyond the reproductive age of the individual, is expected to provide diminishing fitness advantages and, consequently, a tissue-specific limit exists beyond which the effects of subsequent beneficial mutations will not be large enough to overcome random genetic drift. This "drift-barrier hypothesis" has been successful in explaining variation in a variety of traits such as genome size and mutation rate across diverse taxa[19].

Consider as an example the per generation mutation rate: Selection is generally expected to favor reduced mutation rates[20,21] as it reduces the load of deleterious mutations. Current evidence, however, indicates that differences in mutation rates, which can vary over orders of magnitude across different species, are not the results of physiological constraints on DNA-replication fidelity. Instead, mutation rates in different species are the result of a balance between selection and genetic drift as evidenced by their negative correlation with effective population size[22,23].

In case of hierarchical tissues, it is not well understood how suboptimal tissues (with fewer than optimal hierarchical levels) restrain somatic evolution. On the one hand, washing out (characterized by the proliferative disadvantage) is maximized when progenitor cells can only differentiate and never self-renew. On the other hand, minimizing mutation accumulation (characterized by the lifetime divisional load, i.e., the length of the longest cell lineages) requires non-vanishing self-renewal of the progenitors[8]. Crucially, vanishing self-renewal of the progenitors delegates the self-renewal burden to the stem cells, which would make the cell lineages longer. Understanding the organization of real tissues (with less than optimal complexity) requires us to consider this inherent conflict between maximizing washing out, i.e., making progenitor cells more resistant to mutations leading to cancer, and minimizing the accumulation of the same mutations.

Here, we explore the organizational properties of hierarchical tissues that keep the lifetime risk of cancer below a *threshold* value, determined by the "drift-barrier". We show that under general conditions there exists a trade-off between minimizing mutation accumulation and maximizing the proliferative disadvantage of cells. This trade-off provides an explanation for the observed higher division rate of stem cells than what would be expected solely from the minimization of the accumulation of mutations.

## Results

Consider a minimal generic model of hierarchically organized, self-sustaining tissue with cells arranged into $n + 1$ hierarchical

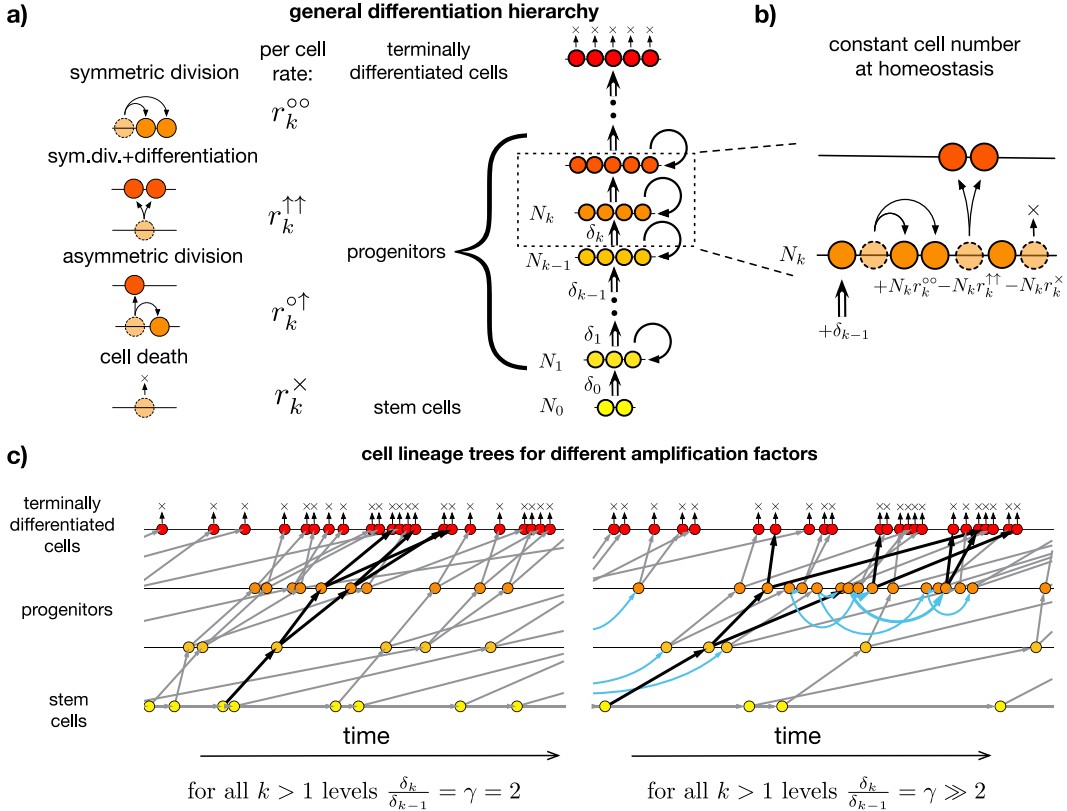

**Fig. 1 Minimal generic model of hierarchically differentiating tissues and corresponding cell lineage trees. a** cells are organized into $n+1$ hierarchical levels based on their differentiation state. The bottom level (level 0) corresponds to tissue-specific stem cells, higher levels represent progressively differentiated progenitor cells, and the top level (level $n$) is comprised of terminally differentiated cells. Four microscopic events can occur with a cell: (i) asymmetric cell division, (ii) symmetric cell division with differentiation, (iv) symmetric cell division without differentiation and (iv) cell death. The number of cells on level $k$ under normal homeostatic conditions is denoted by $N_k$. Under homeostatic conditions each level (except for the terminally differentiated one at the top) provides the next level with newly differentiated cells at a rate $\delta_k$. Terminally differentiated cells at the top of the hierarchy cannot divide and are destined to wear away (i.e., leave the tissue). **b** At progenitor levels $k>0$ in fully developed tissues under homeostatic conditions self-renewal replenishes only a fraction of the cells lost as cells arrive by differentiation from lower levels and, as a result, progenitor cells always have an inherent proliferative disadvantage. **c** Cell lineage trees in "wild type" tissues are shown for two different values of a uniform amplification factor $\gamma_k = \gamma$. Given the same rate of production of terminally differentiated cells (terminal tips of the cell lineage tree) larger values of $\gamma$ correspond to a steeper decline in cell division rates towards lower levels, and slower dividing stem cells with longer progenitor sublineages (cf. the two sublineages highlighted with bold arrows). Longer sublineages for larger values of $\gamma$ are the result of increased self-renewal (i.e., more symmetric cell division events shown in blue) and, equivalently, decreasing proliferative disadvantage of progenitor cells (cf. Eq. (8)).

levels based on their differentiation state[8]: The bottom level (level 0) is comprised of tissue-specific stem cells, while higher levels (levels $k$, where $0 < k < n$) contain progressively more differentiated progenitors, and the top level (level $n$) corresponds to the terminally differentiated cells (Fig. 1a). During tissue homeostasis the stem cell level produces differentiated cells at a rate of $\delta_0$, while the differentiation rates of higher levels (denoted by $\delta_k$ for level $k$) are progressively larger. The increasing tendency of the differentiation rates of the progenitor levels ($0 < k < n$) is specified by the level-specific amplification factors $\gamma_k = \delta_k / \delta_{k-1}$, which relate the differentiation rate of a progenitor level to that of the level below it (cf. Fig. 1b).

**The proliferative disadvantage of cells is determined by the amplification factor.** A differentiation hierarchy $\mathcal{H}$ with $n+1$ levels and $N_0$ stem cells is fully described by the per cell rates of the four microscopic events: symmetric differentiation ($r_k^{\uparrow\uparrow}$), asymmetric differentiation ($r_k^{\circ\uparrow}$), symmetric cell division ($r_k^{\circ\circ}$), and cell death ($r_k^{\times}$) for each level $k$ (Fig. 1a), which together specify the homeostatic cell numbers $N_k$ at each level (Fig. 1b).

The per cell rate of net cell production at level $k$ can be expressed as

$$R_k = r_k^{\uparrow\uparrow} + r_k^{\circ\uparrow} + r_k^{\circ\circ} - r_k^{\times} = A_k - r_k^{\times}, \tag{1}$$

where

$$A_k = r_k^{\uparrow\uparrow} + r_k^{\circ\uparrow} + r_k^{\circ\circ}, \tag{2}$$

the sum of all the three types of cell division rates characterizes the divisional activity of the cells.

Homeostasis, cf. Fig. 1b, implies that on any particular level the number of cells remains constant on average:

$$\delta_{k-1} + N_k r_k^{\circ\circ} - N_k r_k^{\uparrow\uparrow} - N_k r_k^{\times} = \delta_{k-1} - N_k W_k = 0, \tag{3}$$

i.e.,

$$\delta_{k-1} = N_k W_k, \tag{4}$$

where

$$W_k = r_k^{\uparrow\uparrow} - r_k^{\circ\circ} + r_k^{\times} \tag{5}$$

can be identified as the net per cell rate at which cells are depleted ("washed out") from level $k$. Note that because there is no

differentiation toward the stem cell level, $\delta_{-1}$ is formally set to 0 and, therefore, $W_0 = 0$. For all progenitor levels $W_k > 0$.

To derive the relationship between the amplification factor and the strength of "washing out", we first express $\delta_k$, the rate differentiated cells are produced by level $k$ as

$$\delta_k = N_k\left(2r_k^{\uparrow\uparrow} + r_k^{\circ\uparrow}\right) = N_k(R_k + W_k), \qquad (6)$$

where symmetric differentiation events ($\uparrow\uparrow$) produce two differentiated descendants on level $k+1$ and asymmetric differentiation events ($\circ\uparrow$) produce only one descendant on level $k+1$ (the another one on level $k$). Using Eqs. (4) and (9) the amplification factor $\gamma_k$ for the progenitor levels can be expressed as

$$\gamma_k = \frac{\delta_k}{\delta_{k-1}} = \frac{R_k + W_k}{W_k} = 2 + \frac{R_k - W_k}{W_k}. \qquad (7)$$

The proliferative disadvantage of cells at level $k > 0$ can be quantified by the dimensionless ratio of the rate of washing out $W_k$ and the divisional activity $A_k$:

$$\pi_k = \frac{W_k}{A_k} = \frac{W_k}{R_k}\frac{R_k}{A_k} = \frac{1}{\gamma_k - 1}(1 - \varepsilon_k) > 0 \quad \text{for all} \quad k > 0, \quad (8)$$

where

$$\varepsilon_k = \frac{r_k^{\times}}{A_k} \qquad (9)$$

is the death to birth ratio at level $k$.

As we are interested in how tissues of different complexity can restrain somatic evolution, in the following we will omit cell death which can only increase the mutational burden, because a lost cell must always be replaced by cell division. Under such conditions the above expressions simplify to: $A_k = R_k$ and $\varepsilon_k = 0$, from which it follows that $R_k \geq W_k$ and, therefore, $\gamma_k \geq 2$. The minimum of $\gamma_k = 2$ is reached when only symmetric differentiation events ($\uparrow\uparrow$) occur. Finally, the proliferative disadvantage $\pi_k = 1/(\gamma_k - 1)$ becomes a decreasing function of the amplification factor $\gamma_k$ that is maximized at $\gamma_k = 2$ corresponding to progenitors that only divide via symmetric differentiation.

Derényi et al.[8] showed that for a tissue with $N_0$ stem cells that produce a total of $N$ terminally differentiated cells during the tissue's expected lifetime the optimal differentiation hierarchy that minimizes divisional load has $n_{\text{opt}} = \log_2(N/N_0)$ levels and a uniform amplification factor of $\gamma_k = 2$. In such optimal self-sustaining differentiation hierarchies no more than $\log_2(N/N_0) + 2$ cell divisions are sufficient along any cell lineage while, at the same time, the proliferative disadvantage of progenitors is also maximized (cf. Eq. (8)).

However, for suboptimal hierarchies, in particular, ones with $n < n_{\text{opt}}$ levels the amplification factor that minimizes divisional load[8] is $\gamma_k = \gamma^*(n) = (N/N_0)^{1/n} > 2$, while the proliferative disadvantage of progenitors is still maximized by $\gamma_k = 2$. This implies the existence of a trade-off between minimizing divisional load and maximizing proliferative disadvantage. To understand the effect of this trade-off on tissue organization requires developing a quantitative theory of cancer incidence in hierarchical tissues.

**Necessary conditions for cancer.** In healthy tissues the proliferative disadvantage $\pi_k$ of all "wild type" cells except tissue-specific stem cells is strictly positive. As a result, descendants of progenitor cells are inexorably driven toward the terminally differentiated state and eventually lost from the tissue unless they accumulate mutations that lead to a negative proliferative disadvantage at some level of the hierarchy. If the proliferative

disadvantage does become negative at any point in the hierarchy the mutant population will start to grow exponentially.

Mutants with a reduced, but still positive proliferative disadvantage can lead to hyperplasia, which, can be life-threatening. As the conditions for a cell to be able to proliferate into a macroscopically large number or to proliferate exponentially are very similar, we do not distinguish the two, and only focus on uncontrolled growth, which occurs when the proliferative disadvantage becomes negative.

To model the accumulation of mutations we consider driver mutations that each reduces the strength of washing out by the same fraction of the total cell division rate $A_k$, i.e., the rate $\hat{W}_k(d, s)$ at which mutant cells with $d$ driver mutations of strength $s$ are depleted is:

$$\hat{W}_k(d, s) = W_k - d \cdot s \cdot A_k, \qquad (10)$$

for all levels $0 < k < n$. The terminally differentiated level ($k = n$), where only cell loss is assumed to occur, but not cell division, is considered unaffected by driver mutations.

Stem cells ($k = 0$), which must fully self-renew and, as a result, have a proliferative disadvantage of zero, i.e., $\pi_0 = 0$, must be considered separately. Formally, even a single driver mutation, no matter how weak, will, if it is not lost, lead to an exponential, albeit potentially very slow expansion of the stem cell pool. The differentiated descendants of these mutant stem cells, however, will still be at a proliferative disadvantage and will be washed out from higher levels $k > 0$ of the hierarchy, unless a sufficient number of drivers are accumulated to overcome the proliferative disadvantage $\pi_k$.

The critical number $d_{\text{crit}}(k, s)$ of driver mutations of strength $s$ necessary on level $k$ to overcome the proliferative disadvantage $\pi_k$ is the smallest value of $d$ for which $\hat{W}_k(d, s) \leq 0$. This gives

$$d_{\text{crit}}(k, s) = \left\lceil \frac{W_k/A_k}{s} \right\rceil = \left\lceil \frac{\pi_k}{s} \right\rceil = \left\lceil \frac{1}{s(\gamma_k - 1)} \right\rceil, \qquad (11)$$

where $\lceil x \rceil$ denotes the ceiling function, i.e., the smallest integer that is equal to or larger than $x$. Note that the increase in proliferative disadvantage for decreasing amplification factors is reflected in a larger critical number of mutations.

Equation (10), however, does not fully specify the effect of driver mutations, as reduction of $\hat{W}_k(d, s)$ can potentially be achieved in two ways: either by increasing $r_k^{\circ\circ}$ or by decreasing $r_k^{\uparrow\uparrow}$. Of these two possibilities mutations increasing $r_k^{\circ\circ}$ alone can always lead to uncontrolled growth if they are of sufficient strength and number, while mutations decreasing only $r_k^{\uparrow\uparrow}$ require the condition that $r_k^{\circ\circ} > 0$, i.e., $\gamma_k > 2$. Mutations affecting $r_k^{\circ\uparrow}$ cannot lead to exponential growth, but do have an effect on the accumulation rate of mutations.

Here we restrict the discussion to mutations that increase $r_k^{\circ\circ}$, i.e., the case where mutants with $d$ driver mutations exhibit increased self-proliferation:

$$\hat{r}_k^{\circ\circ}(d, s) = r_k^{\circ\circ} + d \cdot s \cdot A_k. \qquad (12)$$

This leads to modified total cell division rates:

$$\hat{A}_k(d, s) = A_k + d \cdot s \cdot A_k \qquad (13)$$

and an increased amplification factor for mutant cells:

$$\hat{\gamma}_k(d, s) = \frac{\hat{A}_k(d, s) + \hat{W}_k(d, s)}{\hat{W}_k(d, s)} = \frac{A_k + W_k}{W_k - d \cdot s \cdot A_k} = \frac{\pi_k + 1}{\pi_k - d \cdot s}, \qquad (14)$$

which diverges as $d$ approaches $d_{\text{crit}}$.

**The probability of accumulating the critical number of driver mutations**. Our aim is to calculate the probability of accumulating the critical number $d_{\mathrm{crit}}(s)$ of driver mutations during the lifetime of a tissue hierarchy $\mathcal{H}$. To do so we make the following assumptions: we assume that driver mutations of strength $s$ occur with probability $\mu$ in each descendant cell following a division event and we consider hierarchy $\mathcal{H}$ described by the number of levels $n$, the amplification factors $\gamma_k$, the homeostatic cell numbers $N_k$, the relative contributions of the two differentiation rates $2r_k^{\uparrow\uparrow}/r_k^{\circ\uparrow}$, and the total rate at which stem cells produce differentiated cells $\delta_0 = (2r_0^{\uparrow\uparrow} + r_0^{\circ\uparrow}) \cdot N_0$ during an expected lifetime $t_{\mathrm{life}}$. Note that the driver mutation rate per cell division $\mu$ is the rate at which mutations that lead to a decreased proliferative disadvantage occur and corresponds to the product of the number of driver genes, the average mutational target size per gene, and the base pair mutation rate per cell division.

To calculate the probability $P_{\mathrm{cancer}}(s, \mu, \mathcal{H}, t_{\mathrm{life}})$ of accumulating the critical number of mutations, we introduce an *efficient mathematical tool* capable of handling lineage trees of hierarchically organized tissues even in the presence of mutations that alter the microscopic properties (e.g., the rates of cellular events) of affected cells and, thereby, alter the structures of the corresponding sublineages. (To make the notations more concise, the $\mu$ and $\mathcal{H}$ dependences of the mathematical functions defined below are not indicated).

The key is to determine the probability $Q_k(d, s, t)$ that a single cell at a non-stem level $0 < k$ that has already acquired $d \leq d_{\mathrm{crit}}(s)$ driver mutations at time $t$ gives rise to a sublineage along which the remaining $d_{\mathrm{crit}}(s) - d$ driver mutations needed to reach criticality are eventually accumulated. By definition $Q_k(d_{\mathrm{crit}}(s), s, t) = 1$ for every level $0 < k < n$, except for the terminal one $k = n$, where $Q_n(d, t) = 0$ for every $d$. For any $0 < k < n$ and $d < d_{\mathrm{crit}}(s)$ in the $(k, d)$ parameter space the probabilities $Q_k(d, s, t)$ can be derived recursively from these boundary conditions (defined at the $k = n$ and $d = d_{\mathrm{crit}}$ boundaries).

Any cell that appears on level $k$ is washed out together with all of its descendants from this level at a rate $\hat{W}_k(d, s)$, which means that after the first appearance of the cell at time $t$ its expected number (including that of its descendants on level $k$) decays exponentially with time $t'$ as $\mathrm{e}^{-\hat{W}_k(d,s)(t'-t)}$. The expected number of times some event with rate $r$ involving this cell or its descendants on level $k$ occurs is then $r \cdot \hat{\tau}_k(d, s, t)$, where

$$\hat{\tau}_k(d, s, t) = \int_t^{t_{\mathrm{life}}} \mathrm{e}^{-\hat{W}_k(d,s)\cdot(t'-t)}\mathrm{d}t' = \frac{1 - \mathrm{e}^{-\hat{W}_k(d,s)\cdot(t_{\mathrm{life}}-t)}}{\hat{W}_k(d, s)} = \frac{\hat{P}_k(d, s, t)}{\hat{W}_k(d, s)} \quad (15)$$

and the survival probability $\hat{P}_k(d, s, t)$, as defined by this equation, is the probability of the cell (and its descendants) not being washed out of level $k$ in the time interval between $t$ and $t_{\mathrm{life}}$.

In particular, the expected number of birth events giving rise to differentiated descendants (i.e., cells on one level higher) at a rate $r_k^{\uparrow} = 2r_k^{\uparrow\uparrow} + r_k^{\circ\uparrow}$ is

$$m_k^{\uparrow}(d, s, t) = r_k^{\uparrow} \cdot \hat{\tau}_k(d, s, t) = \frac{\hat{A}_k(d, s) + \hat{W}_k(d, s)}{\hat{W}_k(d, s)} \quad (16)$$
$$\cdot \hat{P}_k(d, s, t) = \hat{\gamma}_k(d, s) \cdot \hat{P}_k(d, s, t),$$

while the expected number of birth events giving rise to undifferentiated descendants (i.e., cells on the same level) at a

rate $\hat{r}_k^{\circ} = 2\hat{r}_k^{\circ\circ}(d, s) + r_k^{\circ\uparrow}$ is

$$m_k^{\circ}(d, s, t) = \hat{r}_k^{\circ} \cdot \hat{\tau}_k(d, s, t) = \frac{\hat{A}_k(d, s) - \hat{W}_k(d, s)}{\hat{W}_k(d, s)} \quad (17)$$
$$\cdot \hat{P}_k(d, s, t) = [\hat{\gamma}_k(d, s) - 2] \cdot \hat{P}_k(d, s, t).$$

For a cell on level $0 < k < n$ with $d < d_{\mathrm{crit}}(s)$ mutations the probability of giving rise to a lineage that eventually accumulates $d_{\mathrm{crit}}(s)$ mutations can be approximated recursively (starting from the $k = n$ and $d = d_{\mathrm{crit}}$ boundaries) as

$$Q_k(d, s, t) = m_k^{\uparrow}(d, s, t) \cdot [(1 - \mu) \cdot Q_{k+1}(d, s, t) + \mu \cdot Q_{k+1}(d + 1, s, t)] + m_k^{\circ}(d, s, t) \cdot \mu \cdot Q_k(d + 1, s, t), \quad (18)$$

where the terms on the right-hand side correspond to three possibilities: (i) a fraction of $1 - \mu$ of the $m_k^{\uparrow}(d, s, t)$ descendants of the cell on level $k + 1$ acquire no driver mutation, and lead to sublineages with probability $Q_{k+1}(d, s, t)$, (ii) a fraction of $\mu$ of these descendants acquire an additional driver mutation, and lead to sublineages with probability $Q_{k+1}(d + 1, s, t)$, and (iii) a fraction of $\mu$ of the undifferentiated descendants on level $k$ manage to acquire a driver mutation before being washed out, and lead to sublineages with probability $Q_k(d + 1, s, t)$. The recursion gives a small (typically negligible) overestimation of the $Q_k(d, s, t)$ probabilities in three respects. First, it replaces probabilities with expected values, thus, it does not discount the possibility of the simultaneous appearance of critical mutants along the $m_k^{\uparrow}(d, s, t)$ and $m_k^{\uparrow}(d, s, t)$ parallel sublineages. This has a negligible effect as long as $Q_k(d, s, t) \ll 1$, which is the typical case (except when mutants with $d_{\mathrm{crit}}(s) - 1$ mutations are almost critical). Second, the survival probability $\hat{P}_k(d, s, t)$ accounts for the limited time available for a cell (including its descendants) on level $k$, if the cell appears close to the end of the lifetime of the tissue (measured in units of $1/\hat{W}_k(d, s)$), however, the even shorter times available for the sublineages initiated by the $m_k^{\uparrow}(d, s, t)$ and $m_k^{\circ}(d, s, t)$ descendants are not taken into account. The effect of the $\hat{P}_k(d, s, t)$ correction factor is typically very small and this second order correction is even more negligible (again with the exception of the almost critical subcritical mutants). Third, when the critical mutant appears, it may stochastically go extinct before it can establish an exponentially growing population. This, however, is also negligible unless the last driver mutation arrives close to the terminal level $k = n$ and is only slightly critical (i.e., only has a marginally negative $\hat{W}_k(d_{\mathrm{crit}}, s)/\hat{A}_k(d_{\mathrm{crit}}, s)$).

To complete the calculation of $P_{\mathrm{cancer}}(s, \mu, \mathcal{H}, t_{\mathrm{life}})$ we have to consider the accumulation of mutations on the stem cell lineage (i.e., the bottom most line leading to each yellow stem cell on Fig. 1c). To do so, here, we neglect the expansion of the stem cell pool for mutants. This is motivated by the qualitatively different nature of stem cells, which, in contrast to progenitors lack a proliferative disadvantage. The lack of proliferative advantage implies that if stem cells were affected by driver mutations in the same manner as progenitors even a single driver mutation would lead to exponential growth of mutant stem cells. We discuss this assumption in detail below.

The time evolution of the expected number of stem cells $N_0(d, s, t)$ that have acquired $d$ drivers but have not yet given rise to a progenitor sublineage along which the critical number of drivers will have accumulated is given by the following system of

ordinary differential equations:

$$\frac{\partial}{\partial t} N_0(0,s,t) = -r_0^{\circ} \cdot \mu \cdot N_0(0,s,t) - r_0^{\uparrow} \cdot q_0(0,s,t) \cdot N_0(0,s,t) \quad \text{for } d = 0,$$

(19)

$$\frac{\partial}{\partial t} N_0(d,s,t) = r_0^{\circ} \cdot \mu \cdot [N_0(d-1,s,t) - N_0(d,s,t)]$$
$$- r_0^{\uparrow} \cdot q_0(d,s,t) \cdot N_0(d,s,t) \quad \text{for } d > 0,$$

(20)

with initial conditions $N_0(0,s,0) = N_0$ and $N_0(d,s,0) = 0$ for $d > 0$, where $r_0^{\circ} = 2r_0^{\circ\circ} + r_0^{\circ\uparrow}$, $r_0^{\uparrow} = 2r_0^{\uparrow\uparrow} + r_0^{\circ\uparrow}$, and

$$q_0(d,s,t) = (1-\mu) \cdot Q_1(d,s,t) + \mu \cdot Q_1(d+1,s,t) \quad \text{for } d < d_{\text{crit}}(s),$$

(21)

$$q_0(d,s,t) = 1 \quad \text{for } d \geq d_{\text{crit}}(s)$$

(22)

is the probability that a progenitor sublineage descending from a single stem cell accumulates the critical number of mutations.

Using the above the lifetime probability of accumulating $d_{\text{crit}}(s)$ mutations can be expressed as:

$$P_{\text{cancer}}(s, \mu, \mathcal{H}, t_{\text{life}}) = \sum_{d=0}^{\infty} \int_0^{t_{\text{life}}} r_0^{\uparrow} \cdot q_0(d,s,t) \cdot N(d,s,t) \cdot dt$$
$$+ \sum_{k=1}^{n-1} N_k \cdot Q_k(0,s,0),$$

(23)

where the first term describes the probability of accumulating the critical number of mutations from the time $t = 0$ the tissue has fully developed until the end of its expected lifetime $t_{\text{life}}$ and the second term corresponds to the contribution of cells created during tissue development.

Equation (23) can be solved numerically using standard methods and, as shown in Fig. S1, it is in very good agreement with explicit, but extremely time consuming, population dynamics simulations (see Supplementary Note 1). We provide an open-source implementation of both the numerical solution and the explicit simulation used to validate it (see Code availability).

In the following we assume that all amplification factors are equal, i.e., $\gamma_k = \gamma$ and $\hat{\gamma}_k(d,s) = \hat{\gamma}(d,s)$, which also implies that $\pi_k = \pi$ and $d_{\text{crit}}(s) = \lceil \pi/s \rceil = \lceil 1/(s(\gamma-1)) \rceil$ for all progenitor levels $0 < k < n$. The assumption of uniform amplification factors, which corresponds to differentiation rates increasing exponentially along the hierarchy[13,14], is motivated by both mathematical convince and the optimality of identical $\gamma_k$ values in minimizing the lifetime divisional load[8]. Model parameters are summarized in Table S1.

**Quantifying the trade-off between mutation accumulation and proliferative disadvantage.** The above result for the risk of cancer during the lifetime of a tissue hierarchy $\mathcal{H}$ described by the number of levels $n$, the uniform amplification factor $\gamma$, homeostatic cell numbers $N_k$ for $k > 0$ and the total number of stem cell division $\delta_0 = r_0^{\uparrow} \cdot N_0$ has some clear implications: the accumulation of cancer risk during a tissues lifetime, i.e., the first term on the right hand side of Eq. (23), is proportional to $N_0$, increases with increasing tissue life time, and decreases with increasing $n$, because $\delta_0 = \gamma^{1/n}$ and the $Q_0(d,s,t)$ terms describing the probability of accumulating the critical number of mutations along a progenitor lineage also decrease as the divisional load decreases with increasing $n$.

The dependence on the amplification factor $\gamma$, however, is more complicated. As illustrated in Fig. 2b, c, in contrast to the probability of accumulating a fixed number of mutations, the

minimum of the probability of cancer as a function of the amplification factor $\gamma$ is not, in general, close to the value $\gamma^*(n) = (N/N_0)^{1/n}$ that minimizes the lifetime divisional load of the tissue[8]. Instead, the amplification factor $\gamma^*_{\text{cancer}}$ that minimizes the probability of cancer is determined by a trade-off between the proliferative disadvantage along the hierarchy, reflected in increasing $d_{\text{crit}}(s) = \lceil 1/s(\gamma-1) \rceil$ for decreasing $\gamma$ as shown in Fig. 2a, and mutation accumulation, which is minimized near $\gamma^*(n)$, as illustrated by the colored bands in Fig. 2b, c.

The question arises if the trade-off is the result of the stem cells being unaffected by driver mutations. The mathematical description developed here can be readily extended to situations where the drivers do affect the stem cell rates. The results (not detailed here) show that the amplification factor that minimizes the cancer incidence is shifted to higher values (to relieve the cell divisional burden on the stem cells), but a trade-off remains.

To explore the relevance of such a trade-off consider two human tissues, the hierarchical organization of which are best understood: the hematopoietic system, where approximately $N_0 = 10^4$ stem cells[24] produce about $N = 10^{15}$ terminally differentiated cells, and the colon, where approximately $N_0 = 10^8$ stem cells produce $N = 10^{14}$ terminally differentiated cells during a person's lifetime.

For a fully optimal hierarchy with $n_{\text{opt}} = \log_2(N/N_0)$ levels, where $\gamma^*(n) = 2$, the minimum of the lifetime divisional load coincide with maximal proliferative disadvantage along the hierarchy. Based on the above order of magnitude estimates this would require $n_{\text{opt}} \approx 36$ hierarchical levels in blood, while the colon would require $n_{\text{opt}} \approx 20$. In addition, stem cells at the bottom of both hierarchies would only divide twice during an entire lifetime[8].

Detailed modeling of human hematopoiesis has provided estimates of between 17 and 31 hierarchical levels[25], and long-term hematopoetic stem cells are thought to divide at most a few times a year (estimates of every 25–50 weeks[26] and every 2–20 months[27] have been proposed). The colon is organized into millions of crypts, each containing only a few stem cells and selection for driver mutations occurs within single crypts and the number of hierarchical levels in colonic crypts is less clear, but stem cells are known to divide approximately every 4 days[28,29].

From these data it is obvious that neither tissue appears to possess a fully optimal hierarchy, despite evidence that large and rapidly dividing human tissues have evolved increased cancer resistance[6–8]. This observation is consistent with the existence of a "drift-barrier", i.e., that selection can only optimize tissues to the extent that the selective advantage achieved is sufficiently large to overcome genetic drift.

**The organization of hierarchical tissues that have evolved to limit somatic evolution.** To model the existence of a drift-barrier we consider the least complex tissue, i.e., the one with the smallest number of hierarchical levels, that can keep the probability of cancer below a threshold value. We consider the number of stem cells $N_0$ and the number of terminally differentiated cells produced $N$ as fixed by external constraints and vary the rate of driver mutations per cell division $\mu$ and their strength $s$.

We determined the minimum number of levels $n_{\text{drift}}$ and the corresponding uniform amplification factor $\gamma_{\text{drift}}$ necessary to keep the lifetime risk of cancer below the threshold value of 2% for cancers of the hematopoietic system and about 4% for colorectal cancer[30] (see Methods for details). Varying $s$ and $\mu$ in Fig. 3a, b, we show results for the number of levels $n_{\text{drift}}$ and the amplification factor $\gamma_{\text{drift}}$, together with the number of drivers (which is determined by $s$ and $\gamma_{\text{drift}}$, cf. Eq. (11)) and the stem cell division time (determined by $n_{\text{drift}}$, $\gamma_{\text{drift}}$, and $N_0$).

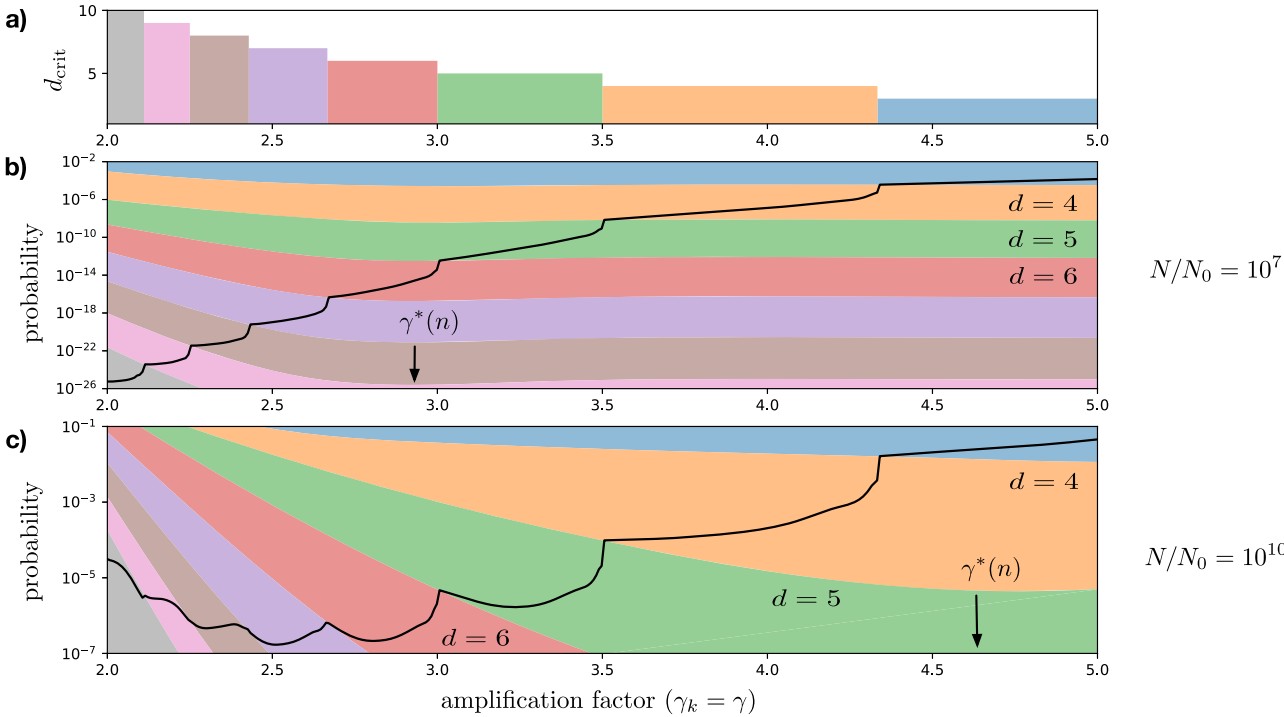

**Fig. 2 Trade-off between mutation accumulation and proliferative disadvantage. a** the number of driver mutations $d(s, \gamma)$ necessary for carcinogenesis for $s = 0.1$ as a function of the uniform amplification factor $\gamma_k = \gamma$. **b, c** Two hierarchies with cells organized into $n + 1$ hierarchical levels, and differing only in the number of terminally differentiated cells produced per stem cell during the lifetime of the tissue, respectively, $N/N_0 = 10^7$ and $N/N_0 = 10^{10}$, are shown. The probability of accumulating a sufficient number of mutations for carcinogenesis to occur, $P_{\text{cancer}}(s, \mu, \mathcal{H}, t_{\text{life}})$ is shown with a thick black line for each. In addition, for both, the range of probabilities for accumulating $d = 3, 4, \dots$ mutations out of $d_{\text{crit}}(s)$ critical mutations are shown with colored bands together with the amplification factor that minimizes divisional load $\gamma_k = \gamma^*(n) = (N/N_0)^{1/n}$. The top of each band corresponds to the probability of accumulating $d - 1$ mutations with strength $s$, while the bottom of each corresponds to accumulating $d$ mutations, cf. Eq. (11). In both plots $N_0 = 1$, $n = 15$, $s = 0.1$ and $\mu = 10^{-6}$.

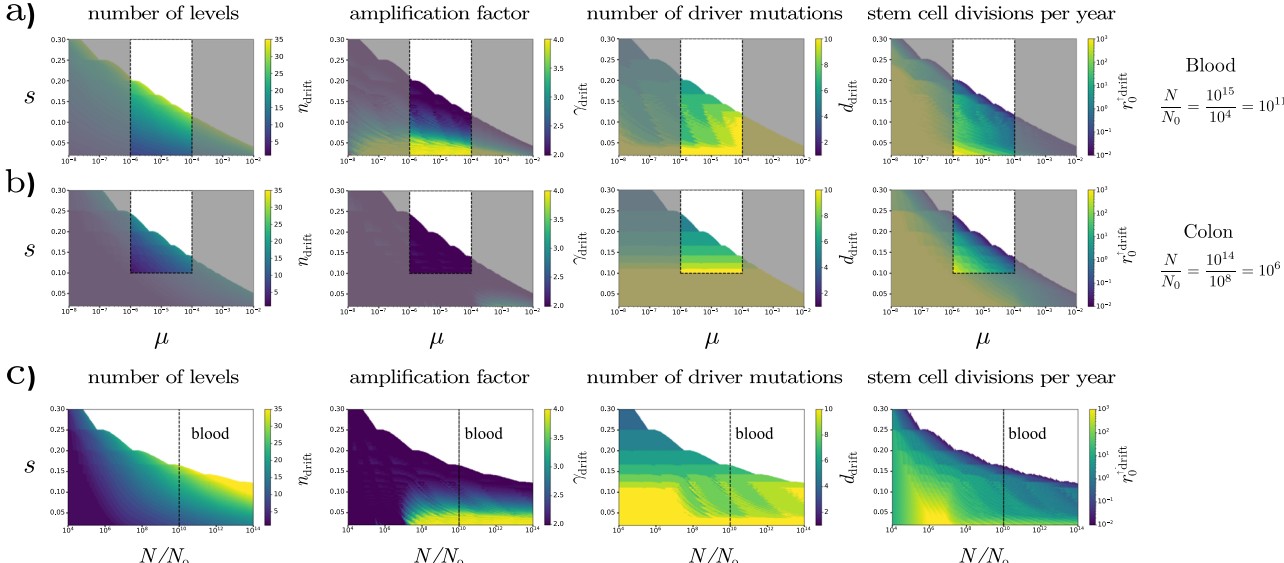

**Fig. 3 The organization of hierarchical tissues that have evolved to limit somatic evolution.** We consider the least complex tissue, i.e., the one with the smallest number of hierarchical levels that can keep the lifetime risk of cancer below a specific value set by the "drift-barrier" (see Methods for details). **a** For the hematopoietic system, where $N_0 = 10^4$ stem cells produce approximately $N = 10^{15}$ mature cells during an individual's lifetime we consider a threshold lifetime risk of about 2%. **b** For the colon, where $N_0 = 10^8$ and $N = 10^{14}$ we consider a threshold of about 4%. Each panel shows the value of a given parameter of the least complex tissue for different values of $s$ and $\mu$. For both tissues, the area delimited by the dashed lines corresponds to the most realistic values for $\mu$ and $s$ based on the literature, as discussed in the main text. **c** To explore tissues with different values of $N/N_0$ we keep $N = 10^{15}$, $\mu = 10^{-5}$ and the maximum acceptable lifetime risk of 2% fixed, but change the ratio $N/N_0$. The dashed line corresponds to the value for the hematopoetic system.

Estimates for the rate of driver mutations per cell division[31–33] vary over $\mu = 10^{-6}$–$10^{-4}$ reflecting potentially tissue specific uncertainty in both the number of mutational targets and the somatic mutation rate per cell division. Recently, Watson et al.[24] have estimated that there are ≥2500 variants that confer moderate to high selective advantages in hematopoietic stem cells, which combined with an approximate mutation rate of $10^{-9}$ per base pair per cell division corresponds to $\mu \approx 2.5 \times 10^{-6}$. For the average selective advantage of driver mutations estimates range from[31] $s \approx 10^{-3}$ to[32,34] $s > 10^{-1}$. For the colon empirical measurements[35] and theoretical arguments suggest that $s < 10^{-1}$ is unlikely, but for blood the entire range of values is plausible, with estimates by Watson et al.[24] indicating that 40% of variants confer moderate to high fitness effects of $s > 0.04$.

The unshaded areas bounded by the dashed lines in Fig. 3a, b show the ranges of $\mu$ and $s$ values consistent with the above estimates. For the hematopoietic system we find that the number of hierarchical levels ranges between $n = 15$ and 30, and the amplification factor between $\gamma = 2$ and 6, broadly consistent with estimates[25] based on available in vivo data. The number of drivers falls between $d = 4$ and 6, while stem cells divide a few times per year. For the colon we find a significantly lower number of levels between $n = 5$ and 15 and an amplification factor of $\gamma = 2$, corresponding to maximal washing out, again consistent with our understanding of the organization of the colorectal epithelium[36–38]. As can be seen when comparing the rightmost panels in Fig. 3a, b, the maximization of washing out in the case of the colon, however, comes at the cost of relegating the burden of cell proliferation to the stem cells, which divide at least an order of magnitude faster as compared to blood, for the same values of $\mu$ and $s$.

## Discussion
Animals have been evolving mechanisms to suppress cancer ever since the origin of multicellularity. The existence of species level adaptations, as exemplified by the near irrelevance of mammalian body size and lifespan to lifelong cancer risk, has been clear for several decades[4,5]. The realization that rapidly renewing tissues of long-lived animals, such as humans, must also have evolved tissue specific protective mechanisms also dates back several decades[9,10]. Evidence for tissue specific adaptations is, however, more recent[6–8].

In the above we calculate the lifetime risk of cancer in a hierarchically differentiating self-renewing tissue taking into account the effects of driver mutations that reduce the proliferative disadvantage of mutants. Using this result we determine the organizational properties of hierarchical tissues that have evolved to limit somatic evolution by keeping the lifetime risk of cancer below a maximum acceptable value. We find that the optimal tissue organization is determined by a trade-off between two competing mechanism, reduced mutation accumulation[8], and increased "washing out" through the progression of increasingly differentiated cell types[15].

We show that such a trade-off exists as long as differentiation hierarchies are not fully optimal in reducing divisional load. This is likely the case in most tissues of most species, as fully optimal tissues require complex hierarchies with a large number of levels incompatible with current empirical evidence[6,8]. Such complex hierarchies are also unlikely to have evolved according to the "drift-barrier" hypothesis[22,23,39] which, in contrast to the view that natural selection fine-tunes every aspect of organisms, predicts that genetic drift, resulting from finite population sizes, can limit the power of selection and constrain the degree to which phenotypes can be optimized by selection.

The trade-off occurs in the tempo of increase of the cell production rate along the differentiation hierarchy, which we parametrize by the amplification factor $\gamma$. The amplification factor corresponds to the ratio of the rate at which adjacent levels produce differentiated cells. Tissues with a smaller amplification factor experience increased mutational burden, however, at the same time exhibit increased washing out, resulting in a trade-off between the two.

We demonstrate that based on the lifetime number of the terminally differentiated cells produced per stem cell, our theoretical description (Fig. 3a, b) provides realistic predictions for the organization of the human hematopoietic system and the epithelial tissue of the colon. In particular, the hematopoietic differentiation hierarchy is predicted to have a relatively larger number of levels with a relatively high amplification factor ensuring low mutational load from cell divisions, in agreement with previous results[25]. The colorectal epithelium, the paradigmatic model of differentiation induced proliferative disadvantage[15,18], in contrast, has a near minimal amplification factor and few differentiation levels ensuring strong washing out and requiring a fast stem cell turnover rate in agreement with experimental data[28,29].

In summary, trade-off theory does not lead to a different optimum, but rather argues that, given the relevant limits of natural selection set by genetic drift, tissues have not evolved to be optimal. The quantitative model developed here provides a general analytical tool for predicting the organization (including the cell differentiation rates and the number of hierarchical levels) of tissues of various sizes ($N_0$ and $N$) based on the rate ($\mu$) and strength ($s$) of driver mutations. Based on these results we demonstrate that under a broad range of parameters characteristic of real tissues, hierarchical structure optimized to the limits of natural selection set by genetic drift is determined by a trade-off between mutation accumulation and the strength of washing out. An immediate consequence of our predictions is the explanation of the surprisingly fast turnover rate of the stems cells of the colonic crypts.

It is, however, important to emphasize that our results only consider the balance between mutation accumulation and washing out resulting from cell differentiation, while keeping other variables fixed. In particular, $N/N_0$, the number of terminally differentiated cells produced per stem cell during the lifetime of the tissue is a constraint of fundamental importance (cf. Figs. 2 and 3). For the two examples considered above, blood and colon, the number of terminally differentiated cells produced during the lifetime of the two tissues is similar (~$N = 10^{15}$ and $10^{14}$, respectively), while the number of cells produced per stem cell differs by several orders of magnitude ($N/N_0 = 10^{11}$ and $10^6$). In fact, the two tissues are markedly different in their physical organization, and this is reflected in the differences in the number of stem cells in each. Blood is replenished in a centralized manner by the bone marrow, while the intestinal epithelium of the colon is renewed in a highly localized manner by a large number of stem cells that reside at the base of a large number of distinct crypts. Understanding the evolutionary and physiological origins of differences in the hierarchical organization of different tissues will require a theory that considers all relevant and evolutionary forces and physiological constraints together.

## Methods
**Calculating the minimum number of levels and the corresponding amplification factor.** For specific values of $N$, $N_0$, $\mu$, and $s$, to determine the minimum number of levels $n$ and the corresponding uniform amplification factor $\gamma$, starting with $n = 1$ we determine the minimum of the lifetime cancer risk (defined by

Eq. (23)) as a function of $\gamma$. If this minimum is above the threshold value of 2% for cancers of the hematopoietic system and about 4% for colorectal cancer[30], we increase $n$ by one, otherwise, we stop the procedure. Fig. S2 (see Supplementary Note 2) shows the robustness of our results to changing the value of the threshold between 0.1% and 10%, while supplementary Fig. S3 (see Supplementary Note 3) explores the effect of driver mutations that do not change the proliferative disadvantage until the critical number is accumulated.

**Reporting summary**. Further information on research design is available in the Nature Research Reporting Summary linked to this article.

## Code availability

Computer code for the numerical solution of Eq. (23) as well explicit simulations used to produce the results presented both in the main text and the supplementary information is available at https://github.com/pentadotddot/TradeOffArticle_supplements[40]. We used python version 3.9.10 with numpy version 1.22.2 and scipy version scipy-1.8.0.

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

## Acknowledgements

G.J.Sz. and M.D. received funding from the European Research Council under the European Union's Horizon 2020 research and innovation program under grant agreement no. 714774.

## Author contributions

M.D., I.D., and G.J.Sz. conceptualized the study and wrote the manuscript. I.D. and G.J.Sz. derived analyitical results. M.D. performed numerical calculations and validation.

## Funding

## Competing interests

The authors decalre no competing interests.
