## [Peer Review File · Nature Communications]

Trade-off between reducing mutational accumulation and increasing commitment to differentiation determines tissue organizationReviewers' Comments:

Reviewer #1:

Remarks to the Author:

Review: Trade-off between reducing mutational load and increasing commitment to differentiation determines tissue organization

In this manuscript under consideration for Nature Communications, Marton Demeter and colleagues theoretically investigate hierarchically organized tissues. In particular they investigate tissue organization, its ability to suppress mutation accumulation and selection by "washing out" positively selected clones through a differentiation process. This manuscript adds to a long list of work on the suppression of somatic evolution in hierarchical tissues.

The authors find a non-trivial trade-off between mutation suppression and cell differentiation. They furthermore show the ability of their model to reproduce cancer incidence data and develop a Bayesian approach to estimate some of the underlying model parameters. Overall the question is interesting and important. However, the manuscript needs to improve in clarity and explanation before publication may be considered. Furthermore, although in theoretical work simplifying assumptions are clearly necessary and warranted, some of the biological assumptions and their justification are probably questionable. Please find a more detailed list below:

The authors argue that for cancer to occur the proliferative disadvantage has to become negative (line 106). This is not necessarily true. One can imagine scenarios where e.g. the self-renewal rate of cells changes across multiple stages of the hierarchy such that in each state the proliferative disadvantage is positive (and cells extinct in the long run), yet the intermittent cancer cell count is already lethal.

In Line 127-129 the authors argue that the assumption of neutral driver mutations until after a critical number of driver mutations is reached is justified by the absence of a histologically discernible pre-malignant phase. I understand this assumption from a theoretical perspective. However, the justification is unfortunately wrong (especially because the authors discuss colon and blood). An adenoma in the colon is the prime example of a premalignant stage that has accumulated some (APC, TP53) driver mutations, but has not yet progressed to an aggressive carcinoma. The same in blood. The impact of clonal hematopoiesis (accumulation of driver mutations with age in otherwise healthy hematopoiesis) is a major branch of research of the last years. Here the authors might want to read. How would allowing intermediate effects of driver mutations influence the conclusions of the authors.

Given that equation (9) has been derived in a different manuscript, the reader needs a bit more detail and understanding on how this equation was derived and what it means. Especially, checking the BioRxiv paper in which it is derived, one can see that the approximation fails at probabilities that are not sufficiently small. How does this influence the conclusions of this manuscript?

The authors say that equation (14) is the main result of the manuscript. They need to expand on the result. Presented as it is, I have a hard time as a reader understanding what is going on. Is there an explicit equation that can be given?

Figure 2 shows an example for the trade-off. Is this trade-off general or only in a certain parameter regime? Are there analytical statements that can be derived?

In the section starting at line 164, the authors discuss colon and blood as an example. They say blood is replenished by 10^4 and colon by 10^8 stem cells. It is important to recognize that these numbers are still heavily discussed and strictly speaking are unknown. Furthermore, colon is organized into millions of crypts, each containing only a few stem cells. Selection for driver mutations occurs within single crypts, a very important distinction that the authors need to discuss at least.

The authors argue that tissue organization can only be optimal until a neutral drift barrier. This makes great sense to me. In population genetics, neutral drift is usually defined as the fixation (or rise in frequency) of neutral mutations. Later in the manuscript (line 185) the authors define the neutral drift barrier as 2% or 4% lifetime cancer risk in blood and colon respectively. To me this definition seems very different compared to what was discussed at the beginning of the manuscript and seems very arbitrary.

In many of the presented examples, the authors use a driver mutation rate of $\mu = 10^{-5}$. Based on a simple order of magnitude calculation, given a size of the genome of 3×10^9 , this would imply $\sim 10^4$ driver mutations per tissue. This number (also clearly unknown) seems way too high. Current large population studies seem to find driver mutations in the order of tens to at most hundreds. Does the conclusion of the authors change, if smaller driver mutation rates are assumed?

More details are needed for the Bayesian fitting approach. What exactly is the definition of cancer incidence in the model? The accumulation of m driver mutations, or the diagnosis of the cancer (as is in practice)? Why was CML & CLL data pooled? In CML we know that a single driver mutation in the hematopoietic stem cells (BCR-ABL fusion gene) is causative. Yet the authors find 5-7 driver mutations with high confidence. This does not make sense really. What happens if the authors analyze CML and CLL separately? The authors should point out that their model cannot explain the incidence of childhood cancers (as this is indeed a different problem of development).

I find it very hard to interpret figure 3. A more detailed explanation on this very loaded yet important figure might help the reader.

1. Watson, C. J. et al. The evolutionary dynamics and fitness landscape of clonal hematopoiesis. *Science* 367, 1449–1454 (2020).

Reviewer #2:

Remarks to the Author:

The manuscript theoretically studies the evolution of the organization of hierarchical tissues that reduce the risk of cancer. In a previous work by the same research group (Ref. [8]), they showed that the hierarchical structure provides a “smart” mechanism to limit the lifetime divisional load, and they can theoretically estimate the optimal design (e.g. compartment number) for minimizing the divisional load. However, it is not very common to see cellular hierarchies achieving fully optimal design in real tissues as predicted by Ref. [8]. In this study, they explore this issue using the same generic model from Ref. [8] and propose a trade-off theory between mutation accumulation and proliferative disadvantage which lead to an evolutionary optimum in the risk of cancer that differs from the optimal design that purely minimizes the lifetime divisional load. They also apply this theory to explain the differences in tissue-specific hierarchical design (e.g. colon and blood).

I think this work provides very interesting new insight into the evolutionary theory of hierarchical organization. In spite of this, major revision should be done.

1. The number of driver mutations required for cancer is crucial, but I cannot follow why Eq. (8) should be the case. As mentioned by the authors, the necessary condition for cancer is the proliferative advantage of mutant cells, but how could this condition relate to Eq. (8)? This should be clarified more clearly. Is the conclusion intimately dependent on the definition of Eq. (8)? What would happen if we use some other necessary condition for cancer?

2. The paper should be reorganized, especially “the probability of accumulating m mutations” section is not well integrated into the whole paper. For example, Eq. (9) shows very little information (even though I agree that it’s a technical basis) and the definition of μ is missing here. Is it possible to summarize the general idea of Ref. [24]?

3. Fig. 1(b) is confusing; what does the horizontal direction mean and how to check the fact that larger values of γ correspond to a steeper decline in cell division rates in this Figure. I don’t get it.

4. The trade-off theory leads to a different type of evolutionary optimum from purely minimizing the lifetime divisional load in Ref [8], which could be used to explain the mismatch between the reality and previous theoretical prediction. However, alternative explanation could be provided. For example, cell number regulation (Ref. 18) does not take into account in this study, but this mechanism also predicts different evolutionary optimum from Ref. [8]. As far as I know quite a few theoretical groups (including this group) have done many theoretical works on the evolution of hierarchical organization based on different model assumptions and different evaluation criterion, by which different optimal solutions against cancer have been predicted. Now the question is how to compare these solutions. This is an issue of model selection. I strongly suggest the authors to discuss this issue in great details.

5. The statistical analysis on age-incidence data should’ve been more formal and serious. Why can the likelihood function be normal distributed? How powerful is it (check e.g. if the likelihood sensitive enough to the estimated parameters) and what kinds of tissue growth models match the assumption of the likelihood (e.g. validate the statistical method using synthetic simulation data)? How is the convergence of the proposed MCMC algorithm?

Reviewer #3:

Remarks to the Author:

The manuscript by Demeter et al, Trade-off between reducing mutational load and increasing commitment to differentiation determines tissue organization, explores how hierarchical tissue organization can act as a cancer defense mechanism in tissue-specific levels. Authors described that hierarchical tissues organization can reduce mutational burden and ‘wash out’ mutations cells via terminal differentiation. The authors generate a model to explore hierarchically organization and cancer risk. Cells in the model can gain a proliferative advantage via somatic mutations, however, ‘washing out’ or differentiation can reduce the proliferative advantage. The number of hierarchies, tempo (amplification factor), and mutation rate varied in the models. Authors propose not to expect to see tissue hierarchies that fully minimize the lifetime divisional load (and hence tumor suppression via reduced somatic mutation accumulation), but set to balance between selection and drift. The authors then validate their predictions and model by providing examples of blood and colon and known estimates on the number of hierarchical levels and estimates of stem cell divisions. From these real-life tissue samples, it appears the tissues have not optimized hierarchy. Lastly, the authors fit their models to age-incidence SEER data, and these results are nicely validated and consistent with their model. The authors conclude there is a trade-off between mutation accumulation and the strength of washing out.

In general, I found the manuscript explores exciting new ideas proposed in the field of cancer evolution. As the authors state, Peto’s Paradox has been studied since the 1970s, but using this theoretical framework to study tissue specific cancer vulnerabilities is relatively new – and for that I think this paper contributes broadly to the field of cancer evolution. However, I did find the paper hard to follow and read. I also had concerns about some assumptions the authors use. Please see below for detailed comments on my concerns.

Major concerns:

I found this paper less accessible to broad readership of the journal. It is a conceptually complicated topic and could use clarifying text edit throughout the manuscript.

Just an example: Line 240: "This is likely the case in most tissues of most species, as fully optimal hierarchies require complex hierarchies with a large number of levels incompatible with current empirical evidence"

Additionally, I find the authors need to clarify their explanations between mutation accumulation and selection (lines 44-49) the way these levels are described in the text sounds like there are defining the same process.

"At the level of mutation accumulation, hierarchical organization can limit the mutational burden of maintaining tissues"; "At the level of selection, even mutations that provide a significant proliferative advantage can be "washed out" as a result of differentiation". Isn't differentiation the end product of hierarchical organization and, hence, the same process? If I am misinterpreting, can the authors expand on this to make it more clear?

Concepts on the balance between selection and genetic drift need to be clearer.

I find the model to be simple, yet sophisticated. Yet the way it is presented I had a hard time following the parameters. I suggest a table with parameters and assumptions to help clarify.

Line 68: Can you give an example of smaller tissues that are much less protected from cancer than larger ones. Does this statement take into account the number of stem cells? As in, you can build a tissue many ways, and it was not the size per-se, but the number of divisions. Muscle/bone are a very large tissues and has a very low risk of cellular transformation.

Another major concern I have is how the manuscript is limited two tissue types: in colon and blood, but then results are stated to be broadly applicable to all tissues. Blood cancer and colon are quite different in the number of drivers necessary and the ecology and function of the tissues. I would like to see in the discussion if these results really can be broadly applicable to different tissues.

Line 33-36: This is an interesting interpretation and deserves an expansion because I don't think this is a consensus in the cancer field – and, from my understanding, still under intense debate. If you are under word limit restriction, I would suggest cutting information on background of multicellular organisms and expanding on this. I was under that impression that the Tomasetti papers, 2015 & 2017, papers are the first to report the "Bad Luck" hypothesis in cancer biology. Which suggests a correlation exists between cancer risk per tissue and lifetime number of stem cell divisions within each tissue, suggesting cancer risk among tissue types can be explained by the accumulation of bad luck mutations (Tomasetti et al., 2017; Tomasetti and Vogelstein, 2015). I believe it was Noble/Hochberg in 2015 that described Peto's Paradox for tissues, and that these tissues get less cancer than expected for stem cell divisions. Expansion on these ideas would help clarify the text.

Line 177" This observation is consistent with the existence of a "drift-barrier" that selection can only optimize tissues to the extent that the selective advantage achieved is sufficiently large to overcome genetic drift.

Authors final conclusions, if I understand them correctly – are that tissues are not optimizing hierarchical structure due to drift – i.e. there is noise in the system. I was surprised to see there is no expansion or clear discussion on why this might be. Authors allude to trade-offs – but what is driving

this?

Minor Edits:

Line 127: spelling of "divers" to "drivers"

Would have liked to see the statistics on cancer in other rapidly dividing tissues, perhaps in the discussion when talking about broad implications above?

Line 130: I would change the phrase 'cooperation' of driver mutations. The accumulation of these break down of the cell cycle circuit system. Cooperation makes the driver mutations intentional.

REVIEWER COMMENTS

Reviewer #1 (Remarks to the Author):

Review: Trade-off between reducing mutational load and increasing commitment to differentiation determines tissue organization

In this manuscript under consideration for Nature Communications, Marton Demeter and colleagues theoretically investigate hierarchically organized tissues. In particular they investigate tissue organization, its ability to suppress mutation accumulation and selection by “washing out” positively selected clones through a differentiation process. This manuscript adds to a long list of work on the suppression of somatic evolution in hierarchical tissues.

The authors find a non-trivial trade-off between mutation suppression and cell differentiation. They furthermore show the ability of their model to reproduce cancer incidence data and develop a Bayesian approach to estimate some of the underlying model parameters. Overall the question is interesting and important.

We would like to thank the reviewer for the positive overall assessment.

However, the manuscript needs to improve in clarity and explanation before publication may be considered.

As detailed below, we have thoroughly revised the manuscript for clarity, completely rewriting several sections.

Furthermore, although in theoretical work simplifying assumptions are clearly necessary and warranted, some of the biological assumptions and their justification are probably questionable. Please find a more detailed list below:

We have constructed and validated a completely new theoretical framework that explicitly models selection and added other modifications to the manuscript that address the points raised.

The authors argue that for cancer to occur the proliferative disadvantage has to become negative (line 106). This is not necessarily true. One can imagine scenarios where e.g. the self-renewal rate of cells changes across multiple stages of the hierarchy such that in each state the proliferative disadvantage is positive (and cells extinct in the long run), yet the intermittent cancer cell count is already lethal.

The reviewer is correct that a reduced, but still positive proliferative disadvantage can lead to potentially significant hyperplasia. As we explain in the “*Necessary conditions for cancer*” subsection of the “*Results*” section, however, the extent to which it will occur depends on the details of cell number

regulation, something that is outside the scope of the present work. We write in the revised version of the manuscript: *“Mutants with a reduced, but still positive proliferative disadvantage can lead to hyperplasia, which, can be life-threatening. As the conditions for a cell to be able to proliferate into a macroscopically large number or to proliferate exponentially are very similar, we do not distinguish the two, and only focus on uncontrolled growth, which occurs when the proliferative disadvantage becomes negative.”*

In Line 127-129 the authors argue that the assumption of neutral driver mutations until after a critical number of driver mutations is reached is justified by the absence of a histologically discernible pre-malignant phase. I understand this assumption from a theoretical perspective. However, the justification is unfortunately wrong (especially because the authors discuss colon and blood). An adenoma in the colon is the prime example of a premalignant stage that has accumulated some (APC, TP53) driver mutations, but has not yet progressed to an aggressive carcinoma. The same in blood. The impact of clonal hematopoiesis (accumulation of driver mutations with age in otherwise healthy hematopoiesis) is a major branch of research of the last years. Here the authors might want to read1. How would allowing intermediate effects of driver mutations influence the conclusions of the authors.

We agree with this important point, and now consider the selective effect of mutations. Please see the first three subsections of the *“Results”* section for details.

Given that equation (9) has been derived in a different manuscript, the reader needs a bit more detail and understanding on how this equation was derived and what it means. Especially, checking the BioRxiv paper in which it is derived, one can see that the approximation fails at probabilities that are not sufficiently small. How does this influence the conclusions of this manuscript?

This equation has been removed as we no longer rely on the results from the BioRxiv manuscript in question.

The authors say that equation (14) is the main result of the manuscript. They need to expand on the result. Presented as it is, I have a hard time as a reader understanding what is going on. Is there an explicit equation that can be given?

This equation has been removed and the presentation of the results improved.

Figure 2 shows an example for the trade-off. Is this trade-off general or only in a certain parameter regime? Are there analytical statements that can be derived?

We were not able to derive explicit analytical results, but we have developed a general and efficient mathematical tool that can handle the strengths of mutations, and extensively explored the parameter regime where a trade-off is present, please see Fig. 3 and supplementary figures S2 and S3 for details.

In the section starting at line 164, the authors discuss colon and blood as an example. They say blood is replenish by 10^4 and colon by 10^8 stem cells. It is important to recognize that these numbers are still heavily discussed and strictly speaking are unknown.

We have added appropriate clarification in the text emphasizing that these are approximate order of magnitude estimates.

Furthermore, colon is organized into millions of crypts, each containing only a few stem cells. Selection for driver mutations occurs within single crypts, a very important distinction that the authors need to discuss at least.

This is indeed an import distinction that we have neglected to explicitly note, we now write: *“The colon is organized into millions of crypts, each containing only a few stem cells and selection for driver mutations occurs within single crypts and the number of hierarchical levels in colonic crypts is less clear, but stem cells are known to divide approximately every 4 days^{29,30}.”*

The authors argue that tissue organization can only be optimal until a neutral drift barrier. This makes great sense to me. In population genetics, neutral drift is usually defined as the fixation (or rise in frequency) of neutral mutations. Later in the manuscript (line 185) the authors define the neutral drift barrier as 2% or 4% lifetime cancer risk in blood and colon respectively. To me this definition seems very different compared to what was discussed at the beginning of the manuscript and seems very arbitrary.

The particular choice of thresholds was based on an order of magnitude estimate of the cancer lifetime incidence. As shown in supplementary figure S2a-i changing this threshold between 0.1%-10% has only a minor effect on tissue organization.

In many of the presented examples, the author use a driver mutation rate of $\mu = 10^{-5}$. Based on a simple order of magnitude calculation, given a size of the genome of 3×10^9 , this would imply $\sim 10^4$ driver mutations per tissue. This number (also clearly unknown) seems way to high. Current large population studies seem to find driver mutations in the order of tens to at most hundreds. Does the conclusion of the authors changes, if smaller driver mutation rates are assumed?

In Fig.3a, Fig.3b and supplementary figures S2 and S3 we show the effect of varying the mutation rate. Tissue organization is not sensitive to the mutation rate and remains largely invariant in the most likely range $10^{-4} \leq \mu \leq 10^{-6}$. We have also revised the text to clarify that the driver mutation rate is the product of the number of driver genes and the average mutational target size, i.e. the number of mutation targets per driver gene that lead to a decreased proliferative disadvantage. We now write: *“Note that the driver mutation rate per cell division μ is the rate at which mutations that lead to a decreased proliferative disadvantage occur and corresponds to the product of the number of driver genes, the average mutational target size per gene, and the base pair mutation rate per cell division.”*

More details are needed for the Bayesian fitting approach. What exactly is the definition of cancer incidence in the model? The accumulation of m driver mutations, or the diagnosis of the cancer (as is in practice)? Why was CML & CLL data pooled? In CML we know that a single driver mutation in the hematopoietic stem cells (BCR-ABL fusion gene) is causative. Yet the authors find 5-7 driver mutations with high confidence. This does not make sense really. What happens if the authors analyze CML and CLL separately? The authors should point out that their model cannot explain the incidence of childhood cancers (as this is indeed a different problem of development).

We have removed the Bayesian fitting approach from the manuscript.

I find it very hard to interpret figure 3. A more detailed explanation on this very loaded yet important figure might help the reader.

We revised both the figure and its caption.

1. Watson, C. J. et al. The evolutionary dynamics and fitness landscape of clonal hematopoiesis. *Science* 367, 1449–1454 (2020).

We would like to thank the reviewer for pointing out this paper. We now cite Watson et al.'s highly informative paper several times in the revised version of the manuscript.

Reviewer #2 (Remarks to the Author):

The manuscript theoretically studies the evolution of the organization of hierarchical tissues that reduce the risk of cancer. In a previous work by the same research group (Ref. [8]), they showed that the hierarchical structure provides a “smart” mechanism to limit the lifetime divisional load, and they can theoretically estimate the optimal design (e.g. compartment number) for minimizing the divisional load. However, it is not very common to see cellular hierarchies achieving fully optimal design in real tissues as predicted by Ref. [8]. In this study, they explore this issue using the same generic model from Ref. [8] and propose a trade-off theory between mutation accumulation and proliferative disadvantage which lead to an evolutionary optimum in the risk of cancer that differs from the optimal design that purely minimizing the lifetime divisional load. They also apply this theory to explain the differences in tissue-specific hierarchical design (e.g. colon and blood).

I think this work provides very interesting new insight into the evolutionary theory of hierarchical organization. In spite of this, major revision should be done.

We would like to thank the reviewer for the positive assessment and accurate summary.

1. The number of driver mutations required for cancer is crucial, but I cannot follow why Eq. (8) should be the case. As mentioned by the authors, the necessary condition for cancer is the proliferative advantage of mutant cells, but how could this condition relate to Eq. (8)? This should be clarified more clearly. Is the conclusion intimately dependent on the definition of Eq. (8)? What would happen if we use some other necessary condition for cancer?

We have completely rewritten this section and revised the mathematical presentation.

2. The paper should be reorganized, especially “the probability of accumulating m mutations” section is not well integrated into the whole paper. For example, Eq. (9) shows very little information (even though I agree that it’s a technical basis) and the definition of μ is missing here. Is it possible to summarize the general idea of Ref. [24]?

We have constructed and validated a new theoretical framework that explicitly takes into account the selective effects of driver mutations and is independent of the results presented in ref. [24].

3. Fig. 1(b) is confusing; what does the horizontal direction mean and how to check the fact that larger values of γ correspond to a steeper decline in cell division rates in this Figure. I don’t get it.

We have revised the figure and its caption, in particular, we explicitly indicate that the horizontal direction corresponds to the passing of time, and use consistent symbols for events across the different panels of the figure.

4. The trade-off theory leads to a different type of evolutionary optima from purely minimizing the lifetime divisional load in Ref [8], which could be used to explain the mismatch between the reality and previous theoretical prediction. However, alternative explanation could be provided. For example, cell number regulation (Ref. 18) does not take into account in this study, but this mechanism also predicts different evolutionary optimum from Ref. [8].

As we clarify in the revised version of the manuscript: *“In summary, trade-off theory does not lead to a different optima in a strict sense, but rather argues that, given the relevant limits of natural selection set by genetic drift, tissues have not evolved to be fully optimal. The quantitative model of the trade-off between mutation accumulation and washing-out developed here provides a general analytical tool for predicting the organization (including the cell differentiation rates and the number of hierarchical levels) of tissues of various sizes (N_0 and N) based on the rate (μ) and strength (s) of driver mutations. Based on these results we demonstrate that under a broad range of parameters characteristic of real tissues, hierarchical structure optimized to the limits of natural selection set by genetic drift is determined by a trade-off between mutation accumulation and the strength of washing-out. An immediate consequence of our predictions is the explanation of the surprisingly fast turnover rate of the stems cells of the colonic crypts.”*

As far as I know quite a few theoretical groups (including this group) have done many theoretical works on the evolution of hierarchical organization based on different model assumptions and different evaluation criterion, by which different optimal solutions against cancer have been predicted. Now the question is how to compare these solutions. This is an issue of model selection. I strongly suggest the authors to discuss this issue in great details.

We have extended the manuscript to include a discussion of the limitations of our results: *“It is, however, important to emphasize that our results only consider the balance between mutation accumulation and washing-out resulting from cell differentiation, while keeping other variables fixed. In particular, N/N_0 , the number of terminally differentiated cells produced per stem cell during the lifetime of the tissue is a constraint of fundamental importance (Figs 2 and 3). For the two examples considered above, blood and colon, the number of terminally differentiated cells produced during the lifetime of the two tissues is similar (approximately 10^{15} and 10^{14}), while the number produced per stem cell differs by orders of magnitude (10^{10} and 10^6). In fact, the two tissues are markedly different in their physical organization, and this is reflected in the differences in the number of stem cells in each. Blood is replenished in a centralized manner by the bone marrow, while the intestinal epithelium of the colon is renewed in a highly localized manner by a large number of stem cells that reside at the base of a large number of distinct crypts. To understand the evolutionary and physiological origins of differences in the hierarchical*

organization of different tissues it will require a theory that considers all relevant and evolutionary forces and physiological constraints together.”

5. The statistical analysis on age-incidence data should've been more formal and serious. Why can the likelihood function be normal distributed? How powerful is it (check e.g. if the likelihood sensitive enough to the estimated parameters) and what kinds of tissue growth models match the assumption of the likelihood (e.g. validate the statistical method using synthetic simulation data)? How is the convergence of the proposed MCMC algorithm?

We have removed the analysis on age-incidence data from the manuscript.

Reviewer #3 (Remarks to the Author):

The manuscript by Demeter et al, Trade-off between reducing mutational load and increasing commitment to differentiation determines tissue organization, explores how hierarchical tissue organization can act as a cancer defense mechanism in tissue-specific levels. Authors described that hierarchical tissues organization can reduce mutational burden and ‘wash out’ mutations cells via terminal differentiation. The authors generate a model to explore hierarchically organization and cancer risk. Cells in the model can gain a proliferative advantage via somatic mutations, however, ‘washing out’ or differentiation can reduce the proliferative advantage. The number of hierarchies, tempo (amplification factor), and mutation rate varied in the models. Authors propose not to expect to see tissue hierarchies that fully minimize the lifetime divisional load (and hence tumor suppression via reduced somatic mutation accumulation), but set to balance between selection and drift.

The authors then validate their predictions and model by providing examples of blood and colon and known estimates on the number of hierarchical levels and estimates of stem cell divisions. From these real-life tissue samples, it appears the tissues have not optimized hierarchy. Lastly, the authors fit their models to age-incidence SEER data, and these results are nicely validated and consistent with their model. The authors conclude there is a trade-off between mutation accumulation and the strength of washing out.

In general, I found the manuscript explores exciting new ideas proposed in the field of cancer evolution. As the authors state, Peto’s Paradox has been studied since the 1970s, but using this theoretical framework to study tissue specific cancer vulnerabilities is relatively new – and for that I think this paper contributes broadly to the field of cancer evolution.

We would like to thank the reviewer for emphasizing potential broad interest and impact of our results.

However, I did find the paper hard to follow and read. I also had concerns about some assumptions the authors use. Please see below for detailed comments on my concerns.

We have extensively revised the manuscript to improve readability and have revisited model assumptions, as detailed below.

Major concerns:

I found this paper less accessible to broad readership of the journal. It is a conceptually complicated topic and could use clarifying text edit throughout the manuscript.

Just an example: Line 240: “This is likely the case in most tissues of most species, as fully optimal hierarchies require complex hierarchies with a large number of levels incompatible with current empirical evidence”

We have revised the manuscript throughout for better readability and improved conceptual clarity.

Additionally, I find the authors need to clarify their explanations between mutation accumulation and selection (lines 44-49) the way these levels are described in the text sounds like there are defining the same process.

“At the level of mutation accumulation, hierarchical organization can limit the mutational burden of maintaining tissues”; “At the level of selection, even mutations that provide a significant proliferative advantage can be “washed out” as a result of differentiation”. Isn’t differentiation the end product of hierarchical organization and, hence, the same process? If I am misinterpreting, can the authors expand on this to make it more clear?

We completely rewrote the introduction and mathematical development of our model and extended our discussion of the ‘drift barrier’.

Concepts on the balance between selection and genetic drift need to be clearer.

We completely rewrote the introduction and mathematical development of our model and extended our discussion of the ‘drift barrier’, emphasizing that the drift barrier in question is at a higher level of selection, i.e. between individuals in contrast to between somatic cells.

I find the model to be simple, yet sophisticated. Yet the way it is presented I had a hard time following the parameters. I suggest a table with parameters and assumptions to help clarify.

We include a table of parameters in the supplementary information.

Line 68: Can you give an example of smaller tissues that are much less protected from cancer than larger ones. Does this statement take into account the number of stem cells? As in, you can build a tissue many ways, and it was not the size per-se, but the number of divisions. Muscle/bone are a very large tissues and has a very low risk of cellular transformation.

We have revised and extended the relevant section along with the rest of the introduction and now provide an example and reference Tomasetti and Vogelstein’s results, which demonstrate that the total lifetime number of cell divisions is a predictor of per tissue lifetime tumour incidence: “*Peto’s paradox for tissues, however, implies that in real tissues we do not in general see optimal hierarchies that reduce cancer incidence to the lowest possible value. This is reflected in Tomasetti and Vogelstein’s⁶ data by the smaller and slower tissues being less protected against cancer than larger ones scaled to the same size (e.g., cancer of the esophagus vs. colorectal cancer).*”

Another major concern I have is how the manuscript is limited two tissue types: in colon and blood, but then results are stated to be broadly applicable to all tissues. Blood cancer and colon are quite different in the number of drivers necessary and the ecology and function of the tissues. I would like to see in the discussion if these results really can be broadly applicable to different tissues.

We agree that it would have been important to consider results for more tissues, unfortunately, however, current knowledge about the hierarchical tissues is very limited and as a result we could not make further meaningful comparisons.

Line 33-36: This is an interesting interpretation and deserves an expansion because I don't think this is a consensus in the cancer field – and, from my understanding, still under intense debate. If you are under word limit restriction, I would suggest cutting information on background of multicellular organisms and expanding on this. I was under that impression that the Tomasetti papers, 2015 & 2017, papers are the first to report the “Bad Luck” hypothesis in cancer biology. Which suggests a correlation exists between cancer risk per tissue and lifetime number of stem cell divisions within each tissue, suggesting cancer risk among tissue types can be explained by the accumulation of bad luck mutations (Tomasetti et al., 2017; Tomasetti and Vogelstein, 2015). I believe it was Noble/Hochberg in 2015 that described Peto's Paradox for tissues, and that these tissues get less cancer than expected for stem cell divisions. Expansion on these ideas would help clarify the text.

We have extended the discussion of this question in the introduction, and cite the paper by Nobel et al.:

“Similar to large and long-lived species, tissues within an individual that are large and rapidly dividing also face potentially higher rates of somatic evolution and as a result higher incidence of tumors, raising the question if tissue specific mechanism to suppress somatic evolution have also evolved? A recent empirical dataset assembled by Tomasetti and Vogelstein⁶ offers key insight to answer this question. The dataset, which gathers lifetime cancer risk and the total number of divisions of healthy self-replicating cells (i.e., stem cells) for 31 different tissues, displays a striking tendency: the dependence of cancer incidence on the number of stem cell divisions is sublinear. In particular, a hundred-fold increase in the number of divisions only results in a ten-fold increase in incidence^{7,8}. As first pointed out by Nobel et al.⁷ this trend supports theoretical predictions⁹⁻¹¹ that tissues with more stem cell divisions (typically larger ones with rapid turnover, e.g., the colon) are relatively less prone to develop cancer, which by analogy we may call Peto's paradox for tissues^{7,8}.”

Line 177” This observation is consistent with the existence of a ‘drift-barrier’ that selection can only optimize tissues to the extent that the selective advantage achieved is sufficiently large to overcome genetic drift.

Authors final conclusions, if I understand them correctly – are that tissues are not optimizing hierarchical structure due to drift – i.e. there is noise in the system. I was surprised to see there is no expansion or clear discussion on why this might be. Authors allude to trade-offs – but what is driving this?

We have revised the discussion of the drift barrier.

Minor Edits:

Line 127: spelling of “divers” to “drivers”

Corrected.

Would have liked to see the statistics on cancer in other rapidly dividing tissues, perhaps in the discussion when talking about broad implications above?

We have removed the analysis on age-incidence data from the manuscript.

Line130: I would change the phrase ‘cooperation’ of driver mutations. The accumulation of these break down of the cell cycle circuit system. Cooperation makes the driver mutations intentional.

Removed.

REVIEWER COMMENTS

Reviewer #2 (Remarks to the Author):

The revised manuscript is greatly improved.

We would like to thank the reviewer for the positive assessment and their comments to date.

The authors chose to remove the statistical analysis on age-incidence data from the manuscript. Even though I asked some questions about this part, I'm actually interested in this part. There is a long list of theoretical work on the mutation suppression in hierarchical structure. However, data validation is rarely presented in previous works. I think the authors can keep this part. Despite of non-standard research paradigm in statistical analysis, discussions about data validation should be more encouraged in this field.

We agree that data validation is an important avenue to explore and plan to pursue this in future work. However, in the present manuscript we have elected to not present the analysis shown in Fig. 4 of the initial submission. The results in question are tangential to the main results of the manuscript and were obtained using the previous mathematical model that assumed driver mutations are neutral until the critical number are accumulated. Thus, to include an updated version the analysis would need to be repeated using the new mathematical model while at the same time addressing the methodological concerns raised.

Reviewer #3 (Remarks to the Author):

The authors have sufficiently addressed all my concerns. The re-write is well organized and clear. I have no major concerns moving forward.

We would like to thank the reviewer for the positive assessment and their comments to date.

I do have a minor comment:

Line 68 – can the author clarify what they mean by “slower tissues” is this a tissue that divides less often?

Corrected.

Reviewers' Comments:

Reviewer #2:

Remarks to the Author:

The revised manuscript is greatly improved.

The authors chose to remove the statistical analysis on age-incidence data from the manuscript. Even though I asked some questions about this part, I'm actually interested in this part. There is a long list of theoretical work on the mutation suppression in hierarchical structure. However, data validation is rarely presented in previous works. I think the authors can keep this part. Despite of non-standard research paradigm in statistical analysis, discussions about data validation should be more encouraged in this field.

Reviewer #3:

Remarks to the Author:

The authors have sufficiently addressed all my concerns. The re-write is well organized and clear. I have no major concerns moving forward.

I do have a minor comment:

Line 68 – can the author clarify what they mean by “slower tissues” is this a tissue that divides less often?